# Stochastic Occupancy Grid Map Prediction in Dynamic Scenes

**Zhanteng Xie**
Department of Mechanical Engineering
Temple University United States
`zhanteng.xie@temple.edu`

**Philip Dames**
Department of Mechanical Engineering
Temple University United States
`pdames@temple.edu`

**Abstract:** This paper presents two variations of a novel stochastic prediction algorithm that enables mobile robots to accurately and robustly predict the future state of complex dynamic scenes. The proposed algorithm uses a variational autoencoder to predict a range of possible future states of the environment. The algorithm takes full advantage of the motion of the robot itself, the motion of dynamic objects, and the geometry of static objects in the scene to improve prediction accuracy. Three simulated and real-world datasets collected by different robot models are used to demonstrate that the proposed algorithm is able to achieve more accurate and robust prediction performance than other prediction algorithms. Furthermore, a predictive uncertainty-aware planner is proposed to demonstrate the effectiveness of the proposed predictor in simulation and real-world navigation experiments. Implementations are open source at https://github.com/TempleRAIL/SOGMP.

**Keywords:** Environment Prediction, Probabilistic Inference, Robot Learning

## 1 Introduction

Autonomous mobile robots are beginning to enter people's lives and try to help us provide different last-mile delivery services, such as moving goods in warehouses or hospitals and assisting grocery shoppers [1–3]. To realize this vision, robots are required to safely and efficiently navigate through complex and dynamic environments that are full of people and/or other robots. The first prerequisite for robots to navigate and perform tasks is to use their sensors to perceive the surrounding environment. This work focuses on the next step, which is to accurately and reliably predict how the surrounding environment will change based on these sensor data. This will allow mobile robots to proactively act to avoid potential future collisions, a key part of autonomous robot navigation.

Environment prediction remains an open problem as the future state of the environment is unknown, complex, and stochastic. Many interesting works have focused on this prediction problem. Traditional object detection and tracking methods [4, 5] use multi-stage procedures, hand-designed features, and explicitly detect and track objects. More recently, deep learning (DL)-based methods that are detection and tracking-free have been able to obtain more accurate predictions [6–11]. Occupancy grid maps (OGMs), the most widely successful spatial representation in robotics, are the most common environment representation in these DL-based methods. This transforms the complex environment prediction problem into an OGM prediction problem, outlined in Figure 1. Since OGMs can be treated as images (both are 2D arrays of data), the multi-step OGM prediction problem can be thought of as a video prediction task, a well-studied problem in machine learning.

The most common technique for OGM prediction uses recurrent neural networks (RNNs), which are widely used in video prediction [12–14]. For example, Ondruska and Posner [6] first propose an RNN-based deep tracking framework to directly track and predict unoccluded OGM states from raw sensor data. Itkina et al. [7] directly adapt PredNet [13] to predict the dynamic OGMs (DOGMas) in urban scenes. Furthermore, Toyungyernsub et al. [8] decouple the static and dynamic OGMs and

7th Conference on Robot Learning (CoRL 2023), Atlanta, USA.

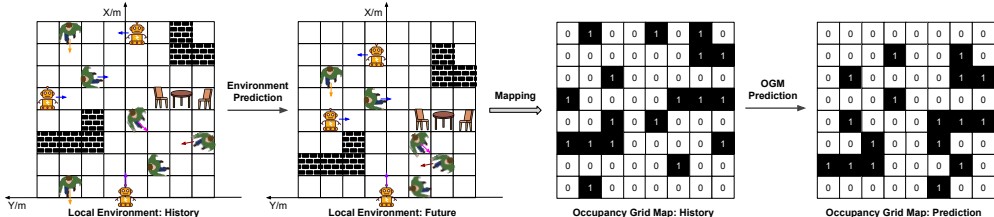

Figure 1: A simple illustration of the OGM prediction problem.

propose a double-prong PredNet to predict occupancy states of the environment. Schreiber et al. [9, 10] embed the ConvLSTM units in the U-Net to capture spatiotemporal information of DOGMas and predict them in the stationary vehicle setting. Lange et al. [11] propose two attention-augmented ConvLSTM networks to capture long-range dependencies and predict future OGMs in the moving vehicle setting. However, these image-based works only focus on improving network architectures and just treat the OGMs as images, assuming their network architectures can implicitly capture useful information from the kinematics and dynamics behind the environment with sufficient good data.

There are other DL-based approaches that explicitly exploit the ego-motion and motion flow of the environment to improve the OGM prediction accuracy. By using input placement and recurrent states shifting to compensate for ego-motion, Schreiber et al. [15] extend their previous image-based works [9, 10] to predict DOGMas in moving vehicle scenarios. By extending the deep tracking framework [6] with a spatial transformer module to compensate for ego-motion, Dequaire et al. [16] propose a gated recurrent unit (GRU)-based network to predict future states in moving vehicle settings. Song et al. [17] propose a GRU-based LiDAR-FlowNet to estimate the forward/backward motion flow between two OGMs and predict future OGMs. Thomas et al. [18] directly encode spatiotemporal information into the world coordinate frame and propose a 3D-2D feedforward architecture to predict futures. By considering the ego-motion and motion flow together, Mohajerin and Rohani [19] first use the geometric image transformation to compensate for ego-motion, and then propose a ConvLSTM-based difference learning architecture to extract the motion difference between consecutive OGMs. However, most of these motion-based works are designed for autonomous vehicles and cannot be directly deployed on mobile robots with limited resources. Furthermore, all the above-described works assume that the environmental state is deterministic and cannot estimate the uncertainty of future states, which we believe is key to helping robots robustly navigate in dynamic environments.

In this paper, we propose two versions of a variational autoencoder (VAE)-based stochastic OGM predictor for resource-limited mobile robots, namely SOGMP and SOGMP++, both of which predict a distribution of possible future states of dynamic scenes. The primary contribution of our approach is that we fully exploit the kinematics and dynamics of the robot and its surrounding objects to improve prediction performance. Specifically, we first develop a simple and effective ego-motion compensation mechanism for robot motion, then utilize a ConvLSTM unit to extract the spatiotemporal motion information of dynamic objects, and finally generate a local environment map to interpret static objects. Another key contribution is that by relaxing the deterministic environment assumption, our proposed approaches are able to provide uncertainty estimates and predict a distribution over future states of the environment with the help of variational inference techniques. We demonstrate the effectiveness of our approaches by using both computer vision metrics and multi-target tracking metrics on three simulated and real-world datasets with different robot models, showing that our proposed predictors have a smaller absolute error, higher structural similarity, higher tracking accuracy than other state-of-the-art approaches, and can provide robust and diverse uncertainty estimates for future OGMs. Note that while all other published works only evaluate the image quality performance of the predicted OGMs, to the best of our knowledge, we are the first to employ a multi-object tracking metric, optimal subpattern assignment metric (OSPA) [20], to more fully evaluate OGM prediction performance in terms of tracking accuracy. In addition, we propose a predictive uncertainty-aware planner by integrating the predicted and uncertain costmaps from our predictor, and demonstrate its superior navigation performance in dynamic simulated and real-world environments.

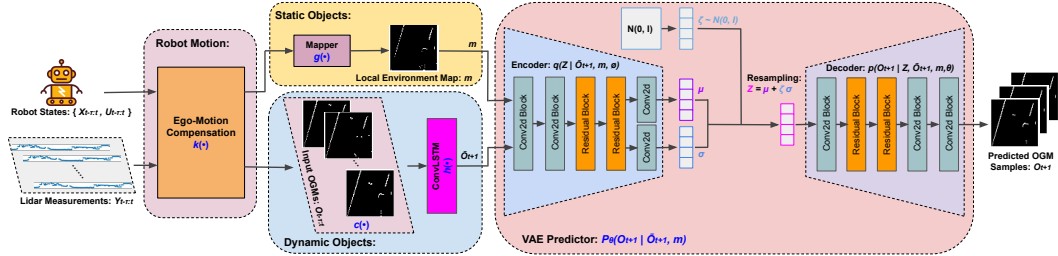

Figure 2: System architecture of the SOGMP++ predictor (SOGMP omits the Static Objects block).

## 2 Stochastic Occupancy Grip Map Predictor

### 2.1 Problem Formulation

In a complex dynamic environment with both static and dynamic obstacles, the robot is equipped with a lidar sensor to sense its surroundings and uses the OGM to represent the environment. We assume each grid cell in the OGM is either occupied or free, *i.e.,* a binary OGM. We assume that the robot is able to obtain relatively accurate estimates of its own pose and velocities from its odometry sensors or other localization algorithms over short periods of time (on the order of $1\,\text{s}$). We denote the pose and control velocity of the robot at time $t$ by $\mathbf{x}_t = [x_t\ y_t\ \theta_t]^T$ and $\mathbf{u}_t = [v_t\ w_t]^T$ respectively. Let $\mathbf{y}_t = [r_t\ b_t]^T$ denote the lidar measurements (range $r$ and bearing $b$) at time $t$, and $\mathbf{o}_t$ denote the OGM at time $t$. Giving a history of $\tau$ lidar measurements $\mathbf{y}_{t-\tau:t}$ and robot states $\{\mathbf{x}_{t-\tau:t}, \mathbf{u}_{t-\tau:t}\}$, the robot needs to predict the future state (*i.e.,* OGM) of the environment $\mathbf{o}_{t+1}$. Then, this environment prediction problem can be formulated as a prediction model:

$$p_\theta(\mathbf{o}_{t+1} \mid \mathbf{y}_{t-\tau:t}, \mathbf{x}_{t-\tau:t}, \mathbf{u}_{t-\tau:t}), \tag{1}$$

where $\theta$ are the model parameters. The goal is to find the optimal $\theta$ to maximize (1).

Note that in this paper, we set $\tau = 10$, the sampling rate is $10\,\text{Hz}$, and limit the physical size of OGMs to $[0, 6.4]\,\text{m}$ along the x axis (forward) and $[-3.2, 3.2]\,\text{m}$ along the y axis (left). We use a cell size of $0.1\,\text{m}$, resulting in $64 \times 64$ OGMs. These settings are consistent with other works on mobile robot navigation [21]. All data $\mathbf{o}, \mathbf{u}, \mathbf{y}$ are represented in the local coordinate frame of the robot.

### 2.2 System Overview

Before describing our proposed SOGMP and SOGMP++ methods, we first briefly discuss image-based prediction methods. The prediction model (1) of image-based approaches is rewritten as

$$p_\theta(\mathbf{o}_{t+1} \mid \mathbf{y}_{t-\tau:t}, \mathbf{x}_{t-\tau:t}, \mathbf{u}_{t-\tau:t}) = f_\theta(\mathbf{o}_{t-\tau:t}), \qquad \mathbf{o}_{t-\tau:t} = c(\mathbf{y}_{t-\tau:t}), \tag{2}$$

where $f_\theta(\cdot)$ is the neural network model, and $c(\cdot)$ is the conversion function to convert the lidar measurements to the binary OGMs in the robot's local coordinate frame. From this image-based model (2), we can easily see that image-based methods explicitly ignore the kinematics and dynamics of the robot and surrounding objects (*i.e.,* assuming they can be implicitly captured by very powerful network architectures and enough good data), and fail to provide a range of possible and reliable OGM predictions (*i.e.,* assuming a deterministic future).

Based on these limitations, we argue that: 1) the future state of the environment explicitly depends on the motion of the robot itself, the motion of dynamic objects, and the state of static objects within the environment; 2) the future state of the environment is stochastic and unknown, and a range of possible future states helps provide robust predictions. With these two assumptions, we fully and explicitly exploit the kinematic and dynamic information of these three different types of objects, utilize the VAE-based network to provide stochastic predictions, and finally propose two novel stochastic OGM predictors (*i.e.,* SOGMP and SOGMP++, shown in Figure 2) to predict the future state of the environment. Since the only difference between the SOGMP and SOGMP++ is that the

SOGMP++ considers the static objects and the SOGMP does not, we mainly describe the SOGMP++ model. The prediction model (1) of our SOGMP++, outlined in Figure 2, can be rewritten as

$$p_\theta(\mathbf{o}_{t+1} \mid \mathbf{y}_{t-\tau:t}, \mathbf{x}_{t-\tau:t}, \mathbf{u}_{t-\tau:t}) = p_\theta(\mathbf{o}_{t+1} \mid \hat{\mathbf{o}}_{t+1}, \mathbf{m}), \tag{3a}$$

$$\hat{\mathbf{o}}_{t+1} = h(\mathbf{o}_{t-\tau:t}), \ \mathbf{o}_{t-\tau:t} = c(\mathbf{y}_{t-\tau:t}^R), \tag{3b}$$

$$\mathbf{m} = g(\mathbf{y}_{t-\tau:t}^R), \tag{3c}$$

$$\mathbf{y}_{t-\tau:t}^R = k(\mathbf{y}_{t-\tau:t}, \mathbf{x}_{t-\tau:t}, \mathbf{u}_{t-\tau:t}). \tag{3d}$$

The VAE predictor module corresponds to (3a). The dynamic object's module corresponds to (3b), where $h(\cdot)$ is the time series data processing function for dynamic objects, $c(\cdot)$ is the conversion function like (2), and $R$ denotes the local coordinate frame of the robot at predicted time $t + n$. The static object's module corresponds to (3c), where $g(\cdot)$ is the occupancy grid mapping function for static objects. Finally, the robot motion module corresponds to (3d), where $k(\cdot)$ is the transformation function for robot motion compensation. Note that this prediction model (3) only predicts future states at the next time step $t + 1$ and is used for training. To predict a multi-step future states at time $t + n$, we can easily utilize the autoregressive mechanism and feed the next-step prediction back $\mathbf{o}_{t+1}$ to our SOGMP/SOGMP++ network (3a) for $n - 1$ time steps to predict the future states at time step $t + n$. Note that the prediction horizon $n$ could theoretically be any time step.

## 2.3 Robot Motion

To account for the robot motion in dynamic scenarios, we propose a simple and effective ego-motion compensation mechanism $k(\cdot)$ to mitigate its dynamic effects and allow the environment dynamics to be consistent in the robot's local coordinate frame. To predict the future OGM state in the robot's local future view, we consider the robot's future ego-motion and transform the observed OGMs $\mathbf{o}_{t-\tau:t}$ to the robot's local coordinate frame at prediction time step $t + n$. So, our proposed ego-motion compensation mechanism can be divided into two steps: robot pose prediction and coordinate transformation. To account for the robot's future ego motion, we first use a constant velocity motion model to predict the robot's future pose. Then, we use a homogeneous transformation matrix to transform the robot poses $\mathbf{x}_{t-\tau:t}$ and lidar measurements $\mathbf{y}_{t-\tau:t}$ to the robot' local future coordinate frame $R$ and compensate for its ego-motion (see Appendix A.1 and Appendix A.2 for details).

## 2.4 Dynamic Objects

Since we use the OGM to represent the environmental state, we first need to implement a conversion function $c(\cdot)$ to convert the compensated lidar measurements $\mathbf{y}_{t-\tau:t}^R$ to the corresponding OGMs $\mathbf{o}_{t-\tau:t}$ by using coordinate to subscript conversion equations (see Appendix A.3 for details).

Tracking and predicting dynamic objects such as pedestrians is the hardest part of environmental prediction in complex dynamic scenes. It requires some techniques to process a set of time series data to capture the motion information. While the traditional particle-based methods [5] require explicitly tracking objects and treat each grid cell as an independent state, recent learning-based methods [7–11, 15–17, 19] prefer to use RNNs to directly process the observed time series OGMs. Based on these trends, we choose the most popular ConvLSTM unit to process the spatiotemporal OGM sequences $\mathbf{o}_{t-\tau:t}$. However, while other works [8, 14] explicitly decouple the dynamic and static/unknown objects and use different networks to process them separately, we argue that the motion of dynamic objects is related to their surroundings, and that explicit disentangling may lose some useful contextual information. For example, pedestrians walking through a narrow corridor are less likely to collide with or pass through surrounding walls. To exploit the useful contextual information between dynamic objects and their surrounding, we directly feed the observed OGMs $\mathbf{o}_{t-\tau:t}$ into a ConvLSTM unit $h(\cdot)$ and implicitly predict the future state $\hat{\mathbf{o}}_{t+1}$ of dynamic objects.

## 2.5 Static Objects

While predicting dynamic objects plays a key role in environmental prediction, paying extra attention to static objects is also important to improve prediction accuracy. The main reason is that the

area occupied by static objects is much larger than that of dynamic objects, as shown in Figure 2, where static objects such as walls are consistently clustered together, while dynamic objects such as pedestrians are sparse and scattered point clusters. Another reason is that static objects contribute to the scene geometry and give a global view of the surroundings, where these static objects maintain their shape and position over time. To account for them, we utilize a local static environment map $\mathbf{m}$ as a prediction for future static objects, which is a key contribution of our work. We generate this local environment map $\mathbf{m}$ using the standard inverse sensor model [22]. This Bayesian approach generates a robust estimate of the local map, where dynamic objects are treated as noise data and removed over time. We speed up this step by implementing a GPU-accelerated OGM mapping algorithm $g(\cdot)$ that parallelizes the independent cell state update operations (see Appendix A.4 for details).

## 2.6 VAE Predictor

If we treat the dynamic prediction function (3b) and the static prediction function (3c) as the prediction step of Bayes filters, then the VAE predictor (3a) is the update step of Bayes filters. It fuses the predicted features of static objects from (3c) and dynamic objects from (3b), and finally predicts a distribution over future states of the environment. A range of possible future environment states allows the robot to capture the uncertainty of the environment and enable risk-aware operational behavior in complex dynamic scenarios. To represent the stochasticity of environment states, we assume that environment states $\mathbf{o}_{t-\tau:t+n}$ are generated by some unobserved, random, latent variables $\mathbf{z}$ that follow a prior distribution $p_\theta(\mathbf{z})$. Then, our VAE predictor model (3a) can be rewritten as

$$p_\theta(\mathbf{o}_{t+1} \mid \hat{\mathbf{o}}_{t+1}, \mathbf{m}) = \int p_\theta(\mathbf{z}) p_\theta(\mathbf{o}_{t+1} \mid \mathbf{z}, \hat{\mathbf{o}}_{t+1}, \mathbf{m}) dz. \tag{4}$$

Since we are unable to directly optimize this marginal likelihood and obtain optimal parameters $\theta$, we use a VAE network to parameterize our prediction model $p_\theta(\mathbf{o}_{t+1} \mid \hat{\mathbf{o}}_{t+1}, \mathbf{m})$, outlined in Figure 2, where the inference network (encoder) parameterized by $\phi$ refers to the variational approximation $q_\phi(\mathbf{z} \mid \hat{\mathbf{o}}_{t+1}, \mathbf{m})$, the generative network (decoder) parameterized by $\theta$ refers to the likelihood $p_\theta(\mathbf{o}_{t+1} \mid \mathbf{z}, \hat{\mathbf{o}}_{t+1}, \mathbf{m})$, and the standard Gaussian distribution $\mathcal{N}(0, 1)$ refers to the prior $p_\theta(\mathbf{z})$. Then, from work [23], we can simply maximize the evidence lower bound (ELBO) loss $\mathcal{L}(\theta, \phi; \mathbf{o}_{t+1})$ to optimize this marginal likelihood and get optimal $\theta$:

$$\mathcal{L}(\theta, \phi; \mathbf{o}_{t+1}) = \mathbb{E}_{q_\phi(\mathbf{z}|\hat{\mathbf{o}}_{t+1}, \mathbf{m})} \left[ \log p_\theta(\mathbf{o}_{t+1} \mid \mathbf{z}, \hat{\mathbf{o}}_{t+1}, \mathbf{m}) \right] - KL\big( q_\phi(\mathbf{z} \mid \hat{\mathbf{o}}_{t+1}, \mathbf{m}) \, \| \, p_\theta(\mathbf{z}) \big). \tag{5}$$

The first term on the right-hand side (RHS) is the expected generative error, describing how well the future environment states can be generated from the latent variable $\mathbf{z}$. The second RHS term is the Kullback–Leibler (KL) divergence, describing how close the variational approximation is to the prior.

Finally, we use mini-batching, Monte-Carlo estimation, and reparameterization tricks to calculate the stochastic gradients of the ELBO (5) [23], and obtain the optimized model parameters $\phi$ and $\theta$. Using them, our VAE predictor can integrate the predicted features $\{\hat{\mathbf{o}}_{t+1}, \mathbf{m}\}$ of dynamic and static objects, and output a probabilistic estimate of the future OGM states with uncertainty awareness.

## 3 Experiments and Results

To demonstrate the prediction performance of our proposed approaches, we first test our algorithms on a simulated dataset and two public real-world sub-datasets from the socially compliant navigation dataset (SCAND) [24]. Second, we characterize the uncertainty of our proposed predictors across different sample sizes and numbers of objects. Finally, we propose a predictive uncertainty-aware planner by using the prediction and uncertainty information from our predictor to demonstrate how it improves robot navigation performance in crowded dynamic simulated/real-world environments. See also Appendix B.1 for experimental evaluation of the run time of different prediction algorithms.

### 3.1 Prediction Results

**Dataset:** To train our networks and baselines, we collected an OGM dataset, called the OGM-Turtlebot2 dataset, using the 3D human-robot interaction simulator with a 0.8 m/s Turtlebot2 robot [25,

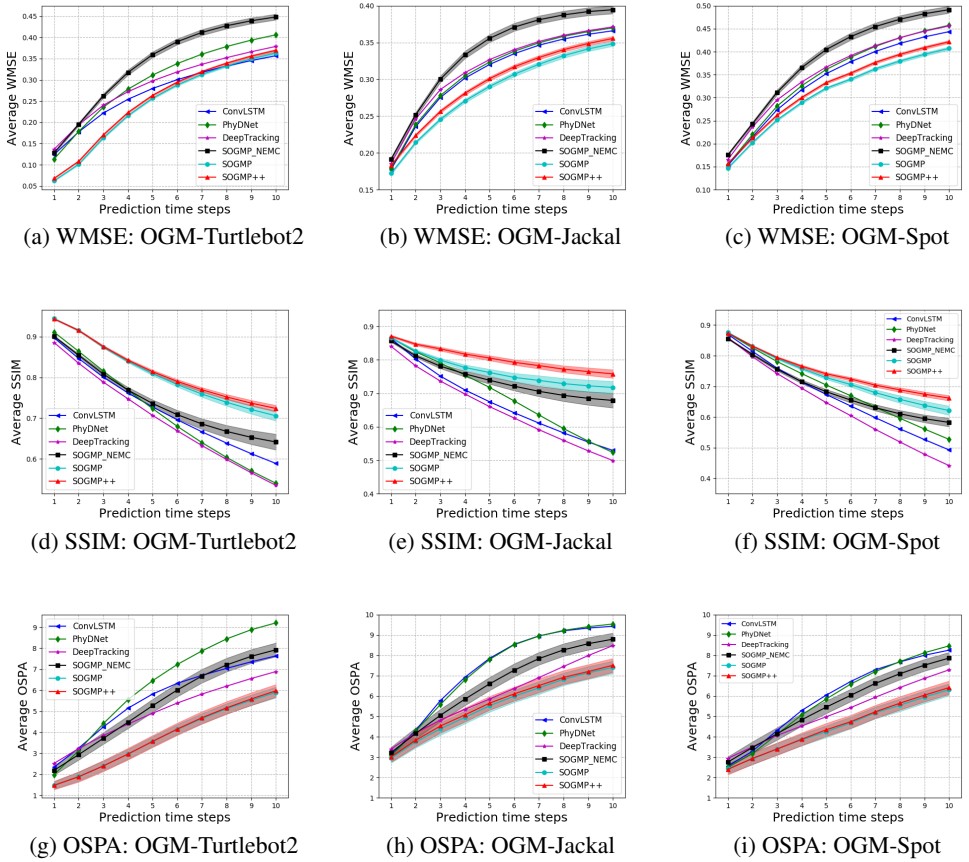

(a) WMSE: OGM-Turtlebot2     (b) WMSE: OGM-Jackal     (c) WMSE: OGM-Spot

(d) SSIM: OGM-Turtlebot2     (e) SSIM: OGM-Jackal     (f) SSIM: OGM-Spot

(g) OSPA: OGM-Turtlebot2     (h) OSPA: OGM-Jackal     (i) OSPA: OGM-Spot

Figure 3: Average WMSE, average SSIM, and average OSPA of 10 different prediction time steps for all tested methods on 3 different datasets. Note that the shadows for SOGMP_NEMC, SOGMP and SOGMP++ approaches are plotted with 95% confidence interval over 32 samples. For all curves for WMSE and OSPA, lower is better. For all curves for SSIM, higher is better.

26]. We collected a total of 94,891 $(\mathbf{x}, \mathbf{u}, \mathbf{y})$ tuples, where 17,000 tuples are used for testing. In addition, to fairly evaluate all networks and examine performance in the real world, we extracted two real-world sub-datasets (*i.e.,* $(\mathbf{x}, \mathbf{u}, \mathbf{y})$ tuples) from the public SCAND [24] dataset: OGM-Jackal dataset (collected by a 2m/s Jackal robot) and OGM-Spot dataset (collected by a 1.6m/s Spot robot). These two real-world datasets are only used for testing. See Appendix A.5 for more details.

**Baselines:** We compare our proposed SOGMP and SOGMP++ algorithms with four DL-based baselines: ConvLSTM [12], DeepTracking [6], PhyDNet [14], and an ablation baseline without the ego-motion compensation module (*i.e.,* SOGMP_NEMC). Note that all networks were implemented by PyTorch framework [27] and trained using the self-collected OGM-Turtlebot2 dataset.

### 3.1.1 Quantitative Results

We evaluate all test OGM predictors on the three datasets from above by evaluating the absolute error, structural similarity, and tracking accuracy (see Appendix A.6 for details). Figure 3 shows the detailed results obtained from these tests, from which we can observe four phenomena. First, our proposed SOGMP predictors with ego-motion compensation achieve significantly better average WMSE, SSIM, and OSPA than the SOGMP_NEMC baseline without ego-motion compensation in all test datasets, which illustrates the effectiveness of ego-motion compensation. Second, the average WMSE of our proposed SOGMP predictors at different prediction time steps is lower than the other image-based baselines in all three test datasets collected by different robots. This shows that the

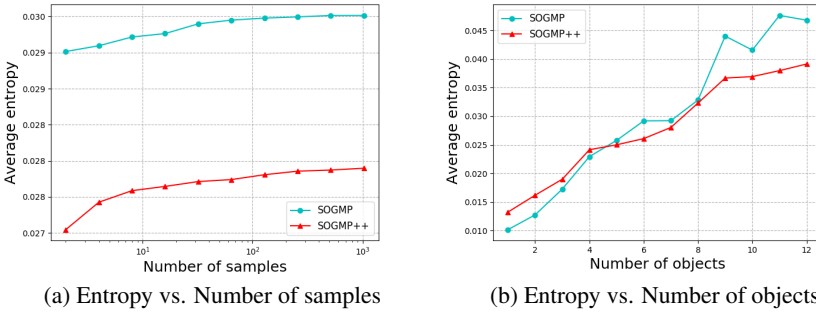

(a) Entropy vs. Number of samples     (b) Entropy vs. Number of objects

Figure 4: Average entropy of our SOGMP and SOGMP++ predictors at 5th prediction time step.

proposed SOGMP predictors utilizing kinematic and dynamic information are able to predict more accurate OGMs (*i.e.,* smaller absolute error) than the state-of-art image-based approaches. Third, while our SOGMP predictors achieve the highest average SSIMs in all test datasets, it is interesting that the average SSIM of our SOGMP++ with a local environment map is significantly higher than that of SOGMP without a local environment map in long-term predictions over multiple time steps. This indicates that the local environment map, which accounts for static objects, helps for longer-term prediction. Finally, the average OSPA errors of our proposed SOGMP predictors are significantly lower than that of the other four baselines, especially in longer prediction time steps. This further demonstrates the preferential performance of our proposed motion-based methods in tracking or predicting environmental states (*i.e.,* localization and cardinality) over other image-based methods. However, the average OSPA errors of our proposed SOGMP and SOGMP++ predictors are almost the same. We believe that this is because static objects are more persistent than dynamic objects, so the static map does not provide significant benefits here. See also Appendix B.2 for examples of predicted OGMs and qualitative analysis.

### 3.1.2  Uncertainty Characterization

One of the biggest differences of our proposed methods compared to other previous baseline works is that they can provide uncertainty estimates. To demonstrate the diversity and consistency of uncertainty estimates, we run two experiments to characterize and analyze the output distribution of our SOGMP/SOGMP++ predictors. One is to show how the entropy of the final probabilistic OGM changes as the number of OGM samples increases, with the hypothesis that it will level off at some value well below that of the entropy of a uniform distribution. The other is to show how the entropy of the final probabilistic OGM changes as the number of objects in the OGM increases, with the hypothesis that it will increase with the number of objects in it. Figure 4 shows the entropy results of these experiments on the OGM-Turtlebot2 dataset, which validates our hypotheses and demonstrates the consistency of our VAE-based predictors. See Appendix A.7 for experimental details.

### 3.2  Navigation Results

We next test the applicability of our proposed methods to practical robotics applications. Figure 5 shows our planner, where we add two new costmaps to the standard ROS [28] navigation stack using the outputs of our SOGMP predictor: one for the predicted scene and one for the uncertainty of the scene. See Appendix A.8 for additional details about the experimental setup.

### 3.2.1  Simulation Results

Using the Turtlebot2 robot in a simulated lobby environment with different crowd densities, as in [25, 26], we compare our proposed predictive uncertainty-aware planner (*i.e.,* DWA-SOMGP-PU) with five navigation baselines: supervised-learning-based CNN [25], deep reinforcement learning-

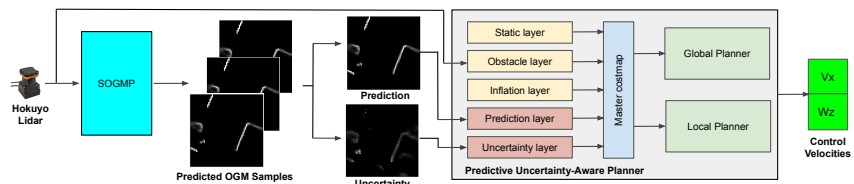

Figure 5: System architecture of the predictive uncertainty-aware navigation planner.

Table 1: Navigation results at different crowd densities

| Environment | Method | Success Rate | Average Time (s) | Average Length (m) | Average Speed (m/s) |
|---|---|---|---|---|---|
| Lobby world, 35 pedestrians | CNN [25] | 0.81 | 14.30 | 5.40 | 0.38 |
| | A1-RC [29] | 0.77 | 16.81 | 6.89 | **0.41** |
| | DWA [30] | 0.82 | 14.18 | 5.15 | 0.36 |
| | DWA-DeepTracking-P | 0.84 | 13.93 | **5.10** | 0.37 |
| | DWA-SOMGP-P | 0.86 | **13.79** | 5.12 | 0.34 |
| | DWA-SOMGP-PU | **0.89** | 14.90 | 5.12 | 0.34 |
| Lobby world, 45 pedestrians | CNN [25] | 0.79 | 16.65 | 5.62 | 0.34 |
| | A1-RC [29] | 0.77 | **14.65** | 6.28 | **0.43** |
| | DWA [30] | 0.77 | 15.39 | 5.16 | 0.34 |
| | DWA-DeepTracking-P | 0.78 | 15.23 | 5.14 | 0.34 |
| | DWA-SOMGP-P | 0.79 | 14.84 | **5.14** | 0.35 |
| | DWA-SOMGP-PU | **0.82** | 15.96 | 5.17 | 0.32 |

based A1-RC [29], model-based DWA [30], and two ablation baselines without the uncertainty costmap (*i.e.,* DWA-DeepTracking-P and DWA-SOMGP-P). Table 1 summarizes these navigation results. As can be seen, our DWA-SOGMP-PU policy has the highest success rate in each crowd size, while having almost the shortest path length. This shows that the prediction costmap from our SOGMP predictor is able to help the traditional DWA planner to provide safer and shorter paths, and combining it with its associated uncertainty costmap can achieve a much better navigation performance than other baselines. It demonstrates the effectiveness of our proposed predictors in helping design safe robot navigation policies in crowded dynamic scenes. See Appendix B.3 for examples of paths planned by each approach and qualitative analysis.

### 3.2.2 Hardware Results

Besides the simulated experiments, we also conduct a real-world experiment to demonstrate the applicability of our DWA-SOGMP-PU policy in the real world. Specifically, we deploy our SOGMP predictor and DWA-SOGMP-PU control policy to a real Turtlebot2 robot with a 2D Hokuyo lidar and an Nvidia Jetson Xavier computer, and let it navigate through a crowded dynamic corridor in a natural condition. From the attached Multimedia, it can be seen that the robot can safely navigate to its goal points among crowded pedestrians without any collision, which demonstrates the real-world effectiveness of our proposed SOGMP predictors and DWA-SOGMP-PU planner.

## 4 Limitations and Future Work

In this paper, we propose two versions of a novel VAE-based OGM prediction algorithm that provides mobile robots with the ability to accurately and robustly predict the future state of crowded dynamic scenes. In addition, we integrate our predictors with the ROS navigation stack to propose a predictive uncertainty-aware planner and demonstrate its effectiveness on the problem of robot crowded navigation. We found that the prediction accuracy of SOGMP predictors was poor when: 1) the robot moves erratically or rotates rapidly; 2) the robot's field of view is occluded by a large obstacle. We also were not able to compare against any of the DL-based ego-motion compensation methods [15–17, 19] as they do not have open-source implementations and our attempts to recreate the results were not successful. Our future work will leverage more accurate robot motion models and correct jump steering angles to overcome these limitations. We will continue to explore how to integrate our uncertainty-aware predictors into learning-based control policies to further improve robot navigation performance in crowded dynamic scenes.

**Acknowledgments**

This work was funded by NSF grant 1830419 and Temple University. This research includes calculations carried out on HPC resources supported in part by the National Science Foundation through major research instrumentation grant number 1625061 and by the US Army Research Laboratory under contract number W911NF-16-2-0189.

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

# A  Additional Implementation Details

## A.1  Robot Pose Prediction

To account for the robot's future ego-motion, we need to predict the future pose of the robot at prediction time step $t + n$. Since the constant velocity motion model is the most widely used motion model for tracking [31] and often outperforms more state-of-the-art methods in general settings [32], we use it as our robot motion model and assume that the robot keeps constant motion in a relatively short period (*i.e.,* less than 1 second). Note that other more suitable robot motion models specific to particular robot models can be used to provide better robot pose predictions. Then, we can easily predict the future pose of the robot $\mathbf{x}_{t+n}$ at prediction time step $t + n$ using the robot's current pose $\mathbf{x}_t$ and velocity $\mathbf{u}_t$:

$$
\begin{bmatrix} x_{t+n} \\ y_{t+n} \\ \theta_{t+n} \end{bmatrix} = \begin{bmatrix} x_t \\ y_t \\ \theta_t \end{bmatrix} + \begin{bmatrix} v_t \cos(\theta_t) \\ v_t \sin(\theta_t) \\ w_t \end{bmatrix} n\Delta t + \begin{bmatrix} \sigma_x \\ \sigma_y \\ \sigma_\theta \end{bmatrix},
\tag{6}
$$

where $\Delta t$ is the sampling interval, $\sigma_{(\cdot)}$ is the Gaussian noise.

## A.2  Coordinate Transformation

To compensate for the ego-motion of the robot, we first use a homogeneous transformation matrix to transform the robot poses $\mathbf{x}_{t-\tau:t}$ to the robot's local future coordinate frame $R$:

$$
\begin{bmatrix} x_{t-\tau:t}^R \\ y_{t-\tau:t}^R \\ 1 \end{bmatrix} = \begin{bmatrix} \cos(\theta_{t+n}) & -\sin(\theta_{t+n}) & x_{t+n} \\ \sin(\theta_{t+n}) & \cos(\theta_{t+n}) & y_{t+n} \\ 0 & 0 & 1 \end{bmatrix}^{-1} \begin{bmatrix} x_{t-\tau:t} \\ y_{t-\tau:t} \\ 1 \end{bmatrix},
\tag{7a}
$$

$$
\theta_{t-\tau:t}^R = \theta_{t-\tau:t} - \theta_{t+n}.
\tag{7b}
$$

Then, by adding these ego-motion displacements, we convert the observed lidar measurements $\mathbf{y}_{t-\tau:t}$ from Polar to Cartesian coordinates:

$$
\mathbf{y}_{t-\tau:t}^R = \begin{bmatrix} x_{t-\tau:t}^z \\ y_{t-\tau:t}^z \end{bmatrix} = \begin{bmatrix} x_{t-\tau:t}^R \\ y_{t-\tau:t}^R \end{bmatrix} + r_{t-\tau:t} \begin{bmatrix} \cos(b_{t-\tau:t} + \theta_{t-\tau:t}^R) \\ \sin(b_{t-\tau:t} + \theta_{t-\tau:t}^R) \end{bmatrix}.
\tag{8}
$$

Finally, we implement the transformation function (3d) and obtain a set of observed lidar measurements $\mathbf{y}_{t-\tau:t}^R$ at the robot's local future coordinate frame $R$. The benefit of ego-motion compensation is that we can treat these lidar measurements from a moving lidar sensor as observations from a stationary lidar sensor at $R$. This significantly reduces the difficulty of OGM predictions and improves accuracy.

## A.3  GPU-accelerated Conversion Function $c(\cdot)$

Algorithm 1 shows the pseudo-code for the GPU-accelerated conversion function $c(\cdot)$ (*i.e.,* (3b)) to convert the lidar measurements to the binary occupancy grid maps.

---
**Algorithm 1:** Converting lidar points to OGMs

---
**Input:** compensated lidar measurements $\mathbf{y}_{t-\tau:t}^R$
**Input:** grid cell size $s$
**Input:** the physical size of the OGMs $S$
**Input:** the lower left corner of the OGMs $(x_0, y_0)$
**Output:** OGMs $\mathbf{o}_{t-\tau:t}$
 1: **initialize:** $\mathbf{o}_{t-\tau:t} = 0$
 2: **for all parallel** beams $z \in \mathbf{y}_{t-\tau:t}^R$ **do**
 3:    $i = \lfloor (x_{t-\tau:t}^z - x_0)/s \rfloor$
 4:    $j = \lfloor (y_{t-\tau:t}^z - y_0)/s \rfloor$
 5:    **if** $i, j \in [0, S/s]$ **then**
 6:       $\mathbf{o}_{t-\tau:t}(i, j) = 1$
 7:    **end if**
 8: **end for**

---

## A.4 GPU-accelerated OGM Mapping Algorithm $g(\cdot)$

Algorithm 2 shows the pseudo code for the GPU-accelerated and parallelized OGM mapping algorithm $g(\cdot)$ [1] (*i.e.,* (3c)) that parallelizes the independent cell state update operation.

Note that $l_i$ is the *log odds* representation of occupancy in the occupancy grid map $m$ [22].

---

**Algorithm 2:** GPU-accelerated OGM mapping

---

**Input:** compensated lidar measurements $\mathbf{y}_{t-\tau:t}^R$
**Output:** local environment map $\mathbf{m}$
1: **for all** time steps $n$ from $t - \tau$ to $t$ **do**
2:     **for all parallel** grid cells $\mathbf{m}_i$ in the perceptual field of $\mathbf{y}_n^R$ **do**
3:         $l_i = l_i + \log \frac{p(\mathbf{m}_i|\mathbf{y}_n^R)}{1-p(\mathbf{m}_i|\mathbf{y}_n^R)} - \log \frac{p(\mathbf{m}_i)}{1-p(\mathbf{m}_i)}$
4:     **end for**
5: **end for**

---

## A.5 Dataset Collections

We collected three OGM datasets to evaluate our proposed prediction algorithm, one self-collected from the simulator (*i.e.,* OGM-Turtlebot2) and two extracted from the real-world dataset SCAND [24] (*i.e.,* OGM-Jackal and OGM-Spot). Note that all three datasets are open source and available online [33].

### A.5.1 OGM-Turtlebot2 Dataset

A simulated Turtlebot2 equipped with a 2D Hokuyo UTM-30LX lidar navigates around an indoor environment with 34 moving pedestrians using random start points and goal points, as shown in Figure 6. The Turtlebot2 uses the dynamic window approach (DWA) planner [30] and has a maximum speed of 0.8 m/s. The Turtlebot2 robot was set up to navigate autonomously in the 3D simulated lobby environment to collect the OGM-Turtlebot2 dataset. The moving pedestrians in the human-robot interaction Gazebo simulator [25, 26] are driven by a microscopic pedestrian crowd simulation library, called the PEDSIM, which uses the social forces model [34, 35] to guide the motion of individual pedestrians:

$$\mathbf{F_p} = \mathbf{F_p^{des}} + \mathbf{F_p^{obs}} + \mathbf{F_p^{per}} + \mathbf{F_p^{rob}}, \tag{9}$$

where $\mathbf{F_p}$ is the resultant force that determines the motion of a pedestrian; $\mathbf{F_p^{des}}$ pulls a pedestrian towards a destination; $\mathbf{F_p^{obs}}$ pushes a pedestrian away from static obstacles; $\mathbf{F_p^{per}}$ models interactions with other pedestrians (*e.g.,* collision avoidance or grouping); and $\mathbf{F_p^{rob}}$ pushes pedestrians away from the robot, modeling the way people would naturally avoid collisions and thereby allowing our control policy to learn this behavior. More details can be found in Xie and Dames [26].

We collected the robot states $\{\mathbf{x}, \mathbf{u}\}$ and raw lidar measurements $\mathbf{y}$ at a sampling rate of 10 Hz. We collected a total of 94,891 $(\mathbf{x}, \mathbf{u}, \mathbf{y})$ tuples, dividing this into three separate subsets for training (67,000 tuples), validation during training (10,891 tuples), and final testing (17,000 tuples).

### A.5.2 OGM-Jackal and OGM-Spot Datasets

Figure 7 shows the real-world outdoor environment at UT Austin and the Jackal robot used to collect the raw SCAND dataset to construct the OGM-Jackal dataset. Figure 8 shows the real-world indoor environment at UT Austin and the Spot robot used to collect the raw SCAND dataset to construct the OGM-Spot dataset. Note that the SCAND dataset was collected by humans manually operating the Jackal robot and the Spot robot around the indoor/outdoor environments at UT Austin. More details can be found in Karnan et al. [24].

---

[1] https://github.com/TempleRAIL/occupancy_grid_mapping_torch

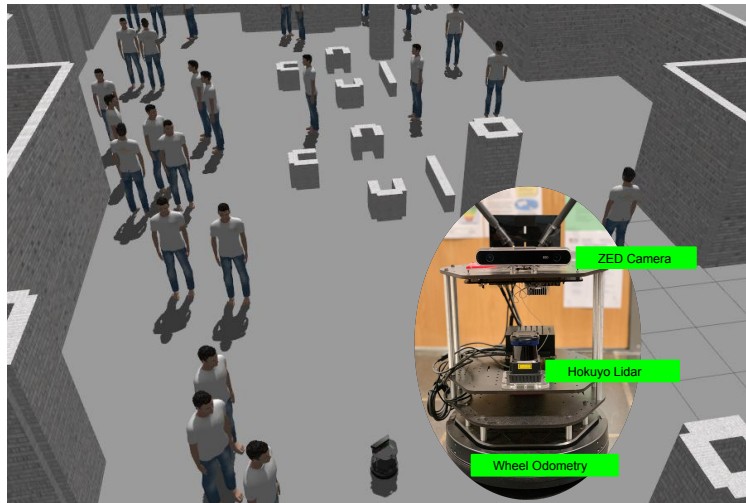

Figure 6: Gazebo simulated environment, where the Turtlebot2 robot was used to collect the OGM-Turtlebot2 dataset.

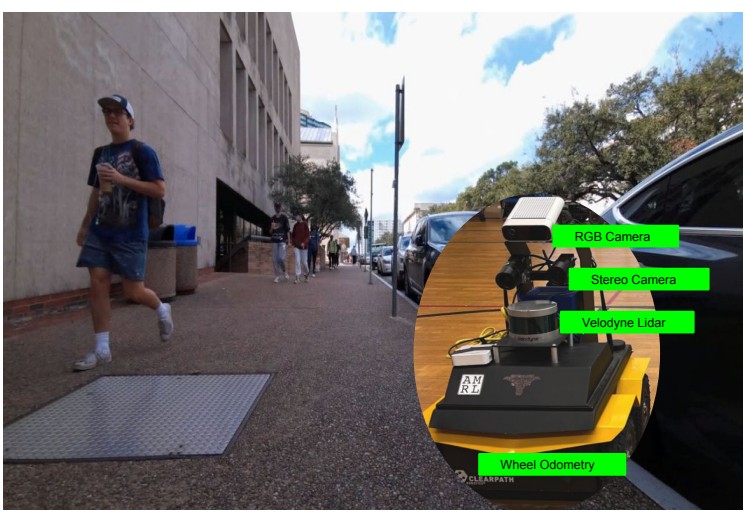

Figure 7: Outdoor environment at UT Austin, where the Jackal robot was used to collect the OGM-Jackal dataset.

### A.6 Experiment Details for OGM Prediction

### A.6.1 Evaluation Metrics

To comprehensively evaluate the performance of OGM predictors, we define the predicted OGM as $\bar{\mathbf{o}}$ and the ground truth OGM as $\mathbf{o}$, and use the following three metrics:

- **Weighted mean square error (WMSE) [36]**:

$$WMSE = \frac{\sum_{i=1}^{N} w_i \left( \bar{\mathbf{o}}_i - \mathbf{o}_i \right)^2}{\sum_{i=1}^{N} w_i}, \tag{10}$$

where $N$ is the number of cells in the OGM, and $w_i$ is the weight for the cell $i$ in the OGM, calculated by the median frequency balancing method [37]. This metric is used to evaluate the weighted absolute errors (balancing the imbalance in the percentage of occupied and free cells) between the predicted OGM and its corresponding ground truth OGM, describing the predicted quality of single OGM cell.

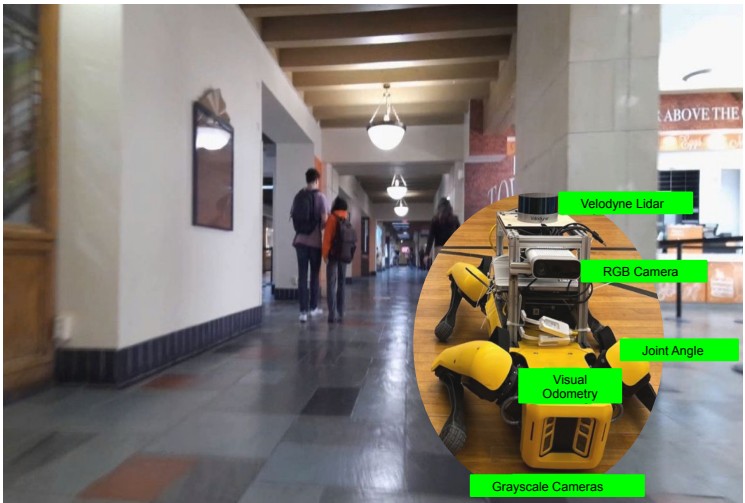

Figure 8: Indoor environment at UT Austin, where the Spot robot was used to collect the OGM-Spot dataset.

- **Structural similarity index measure (SSIM)** [38]:

$$SSIM = \frac{\left(2\mu_{\bar{\mathbf{o}}}\mu_{\mathbf{o}} + C_1\right)\left(2\delta_{\bar{\mathbf{o}}\mathbf{o}} + C_2\right)}{\left(\mu_{\bar{\mathbf{o}}}^2 + \mu_{\mathbf{o}}^2 + C_1\right)\left(\delta_{\bar{\mathbf{o}}}^2 + \delta_{\mathbf{o}}^2 + C_2\right)}, \tag{11}$$

where $\mu_{(\cdot)}$ and $\delta_{(\cdot)}$ denote the mean and variance/covariance, respectively, and $C_{(\cdot)}$ denotes constant parameters to avoid instability. We use $C_1 = 1\mathrm{e}{-4}$ and $C_2 = 9\mathrm{e}{-4}$. This metric is used to evaluate the structural similarity between the predicted OGM and its corresponding ground truth OGM, describing the predicted quality of the scene geometry.

- **Optimal subpattern assignment metric (OSPA)** [20]:

$$OSPA = \left(\frac{1}{n}\min_{\pi \in \Pi_n}\sum_{1}^{m} d_c(\bar{\mathbf{o}}_i, \mathbf{o}_{\pi(i)})^p + c^p(n - m)\right)^{\frac{1}{p}}, \tag{12}$$

where $c$ is the cutoff distance, $p$ is the norm associated to distance, $d_c(\bar{\mathbf{o}}, \mathbf{o}) = \min(c, \|\bar{\mathbf{o}} - \mathbf{o}\|)$, and $\Pi_n$ is the set of permutations of $\{1, 2, ..., n\}$. We use $c = 10$ (*i.e.,* 1 m) and $p = 1$. This metric is used to evaluate the target tracking accuracy between the predicted OGM and its corresponding ground truth OGM, describing the predicted quality of multi-target localization and assignment.

It is worth noting that while other OGM prediction works [7–11, 15–17, 19] only use the computer vision metrics (*e.g.,* MSE, F1 Score, and SSIM) to evaluate the quality of predicted OGMs, we are the first to evaluate the predicted OGMs from the perspective of multi-target tracking (*i.e.,* OSPA). We believe that since it takes into account multi-target localization error and cardinality error, it can give a more accurate and comprehensive evaluation than only evaluating the image quality.

### A.6.2 Evaluation Pipeline for Calculating OSPA Error on OGMs

This evaluation pipeline about how to extract targets from OGMs to calculate their OSPA errors is shown in Figure 9. First, we binarize the predicted OGMs with an occupancy threshold $p_{\mathrm{free}} = 0.3$, which is set by referencing the `occ_thresh` default parameter of 0.25 from the `gmapping` ROS package. Second, we use the density-based spatial clustering of applications with noise (DBSCAN) [39] algorithm to cluster the obstacle points in the OGMs. Finally, we use the mean position of each cluster as the target to calculate the OSPA error (with cutoff distance 10 cells, or 1 m). Note, we get the ground truth target by applying the same process on the ground truth OGMs.

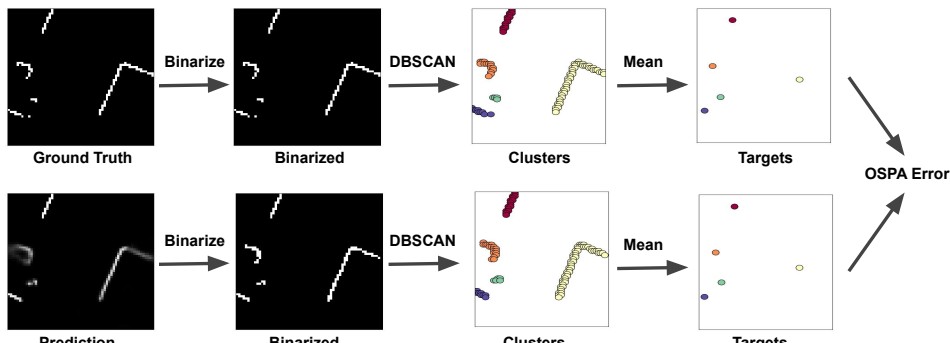

Figure 9: Evaluation pipeline for calculating OSPA error on predicted OGMs.

## A.7 Experiment Details for Uncertainty Characterization

### A.7.1 Evaluation Metrics

To comprehensively characterize the uncertainty information of our SOGMP and SOGMP++ predictors, we define the predicted OGM as $\bar{\mathbf{o}}$ and the ground truth OGM as $\mathbf{o}$ and use the Shannon entropy [40] as the metric:

$$H(\bar{\mathbf{o}}) = \frac{1}{N} \sum_{i=1}^{N} [\bar{\mathbf{o}}_\mathbf{i} \log \bar{\mathbf{o}}_\mathbf{i} + (1 - \bar{\mathbf{o}}_\mathbf{i}) \log(1 - \bar{\mathbf{o}}_\mathbf{i})], \tag{13}$$

where $N$ is the number of cells in the predicted OGM $\bar{\mathbf{o}}$, and $\bar{\mathbf{o}}_\mathbf{i}$ is the value of the cell $i$ in the predicted OGM $\bar{\mathbf{o}}$.

### A.7.2 Experiment Setup

Since our SOGMP/SOGMP++ network predicts a bunch of binary OGM samples (*i.e.,* cell value is 0 or 1) rather than a probabilistic OGM, we first combine these binary OGM samples to create a single probabilistic OGM and then use Shannon entropy to characterize its uncertainty. Note that the number of samples we draw can be scaled according to the robot's available computational resources. Before we conduct our uncertainty experiments, we first compute the number of objects in each input sequence of the OGM-Turtlebot test dataset at the 5th prediction time step using the evaluation pipeline shown in A.6.2, where we classify these input sequences into 12 categories according to the number of objects in them. Then, we randomly select 20 input sequences for each number of objects (*i.e.,* from 1 to 12) and generate a total of 1,024 OGM samples for each input sequence at the 5th prediction time step.

To analyze the relationship between the entropy of the predicted OGM and its sample size, we use all selected test sequences and calculate the average entropy over the number of samples growing as an exponential power of 2. To analyze the relationship between the entropy of the predicted OGM and the number of objects in it, we first use 1,024 OGM samples for each input sequence to generate the final probabilistic OGM and then calculate the average entropy over the number of objects from 1 to 12.

## A.8 Experiment Details for Robot Navigation

### A.8.1 Evaluation Metrics

To comprehensively evaluate the performance of navigation control policies, we use the following four metrics from [25, 26]:

- **Success rate**: the fraction of collision-free trials.
- **Average time**: the average travel time of trials.

- **Average length**: the average trajectory length of trials.
- **Average speed**: the average speed during trials.

### A.8.2 Experiment Setup

For the robot navigation experiments, we use the Turtlebot2 robot with a maximum speed of 0.5 m/s, equipped with a Hokuyo UTM-30LX lidar and an NVIDIA Jetson AVG Xavier embedded computer. Considering the computational resources of the Turtlebot2 robot, we use the SOGMP predictor to generate 8 predicted OGM samples at the 6th prediction time step (*i.e.,* 0.6 s). Based on these 8 predicted OGM samples, we generate a prediction map (mean) and an uncertainty map (standard deviation), which are used to generate the prediction costmap layer and uncertainty costmap layer respectively for our predictive uncertainty-aware planner (*i.e.,* DWA-SOGMP-PU), as shown in Figure 5. Note that each costmap grid cell has an initial constant cost, and we map each occupied grid cell of the prediction costmap and uncertainty costmap to a Gaussian obstacle value rather than a "lethal" obstacle value. This is because the predicted obstacles and uncertainty regions are not real obstacle spaces.

# B    Additional Results

## B.1    Inference Speed Results

Before we focus on quantitative results on the quality of these OGM predictions, we first talk about the inference speed and model size of these predictors. This is because robots are resource-limited, and smaller model sizes and faster inference speeds mean robots have a faster reaction time to face and handle dangerous situations in complex dynamic scenarios.

Table 2 summarizes the inference speed and model size of six predictors tested on an NVIDIA Jetson TX2 embedded computer equipped with a 256-core NVIDIA Pascal @ 1300MHz GPU. We can see that although DeepTracking [6] has the smallest model size, our proposed SOGMP models are about 1.4 times smaller than the ConvLSTM [12] and 4 times smaller than the PhyDNet [14], and their inference speed is the fastest (up to 24 FPS).

Table 2: Inference Speed and Model Size

| Models | ConvLSTM [12] | PhyDNet [14] | DeepTracking [6] | SOGMP_NEMC | SOGMP | SOGMP++ |
|---|---|---|---|---|---|---|
| **FPS** | 2.95 | 4.66 | 5.32 | **24.83** | 23.29 | 10.68 |
| **# of Params** | 12.44 M | 37.17 M | **0.95 M** | 8.84 M | 8.84 M | 8.85 M |

## B.2    Qualitative Prediction Results

Figure 10, Figure 11, Figure 12, and the accompanying Multimedia illustrate the future OGM predictions generated by our proposed predictors and the baselines. We observe two interesting phenomena. First, the image-based baselines, especially the PhyDNet, generate blurry future predictions after 5 time steps, with only blurred shapes of static objects (*i.e.,* walls) and missing dynamic objects (*i.e.,* pedestrians). We believe that this is because these three baselines are deterministic models that use less expressive network architectures, only treat time series OGMs as images/video, and cannot capture and utilize the kinematics and dynamics of the robot itself, dynamic objects, and static objects. Second, the SOGMP++ with a local environment map has a sharper and more accurate surrounding scene geometry (*i.e.,* right walls) than the SOGMP without a local environment map. This difference indicates that the local environment map for static objects is beneficial and plays a key role in predicting surrounding scene geometry.

In addition, Figure 13 shows a diverse set of prediction samples from our SOGMP++ predictor. The red bounding boxes highlight the ability of our SOMGP++ predictor to provide diverse and plausible potential future predictions for stochastic and dynamic environments.

## B.3    Qualitative Navigation Results

Figure 14 shows the difference of nominal paths and costmaps generated by three different planners in the simulated lobby environment. The default DWA planner [30] only cares about the current state of the environment and generates a costmap based on the perceived obstacles. The predictive DWA planner (*i.e.,* DWA-SOGMP-P) using the prediction map of our proposed SOGMP predictor can generate a costmap with predicted obstacles. The predictive uncertainty-aware DWA planner (*i.e.,* DWA-SOGMP-PU) using both the prediction map and uncertainty map of our proposed SOGMP predictor can generate a safer costmap with predicted obstacles and uncertainty regions. These additional predicted obstacles and uncertainty regions of our proposed DWA-SOGMP-PU planner enable the robot to follow safer nominal paths and reduce collisions with obstacles, especially moving pedestrians. See the accompanying Multimedia for a detailed navigation demonstration.

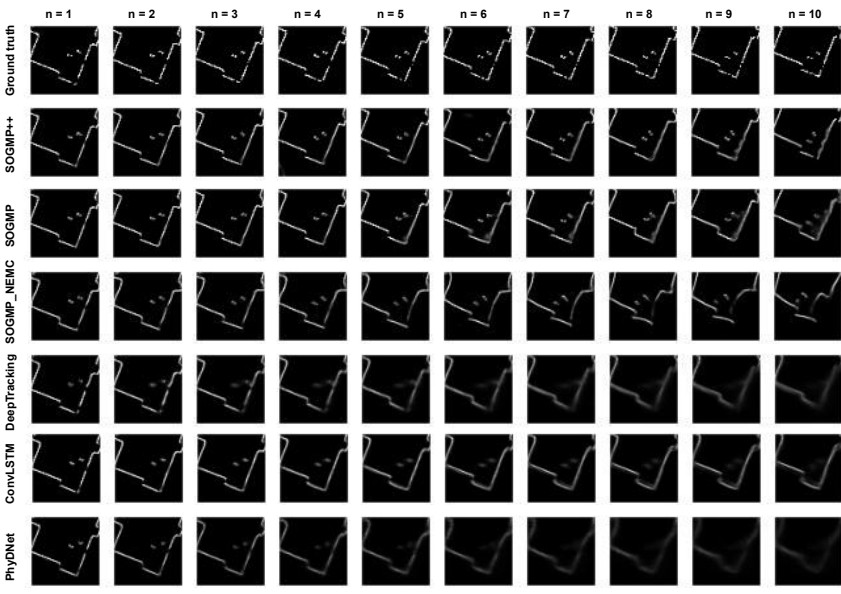

Figure 10: A prediction showcase of the six predictors tested on the OGM-Turtlebot2 dataset over the prediction horizon. The black area is the free space and the white area is the occupied space.

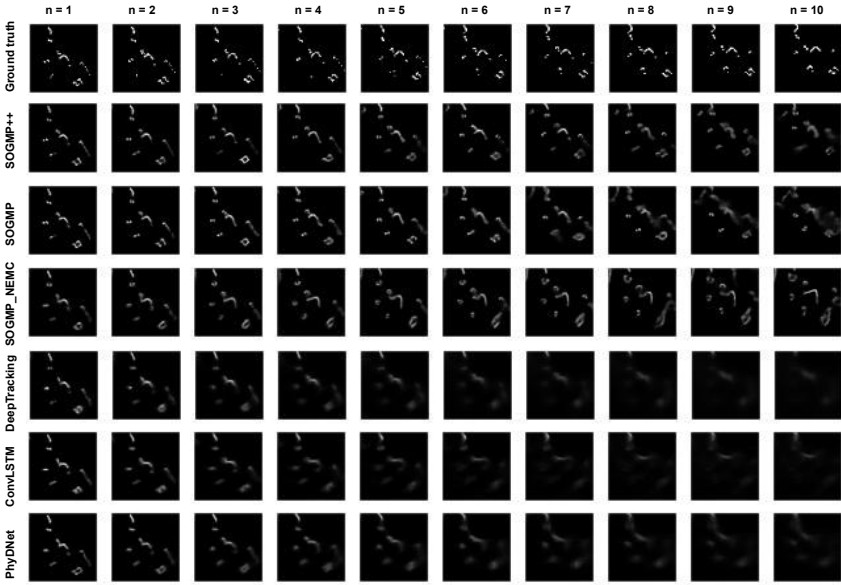

Figure 11: A prediction showcase of the six predictors tested on the OGM-Turtlebot2 dataset over the prediction horizon. The black area is the free space and the white area is the occupied space.

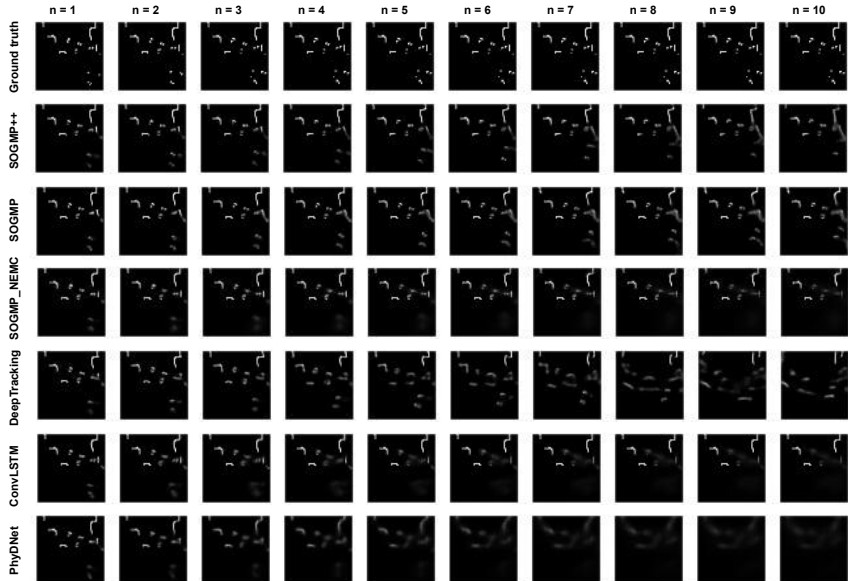

Figure 12: A prediction showcase of the six predictors tested on the OGM-Turtlebot2 dataset over the prediction horizon. The black area is the free space and the white area is the occupied space.



Figure 13: A diverse set of prediction samples of our SOGMP++ predictor tested on the OGM-Turtlebot2 dataset at the 5th prediction timestep. The black area is the free space and the white area is the occupied space. The red bounding boxes show multiple possible predictions.

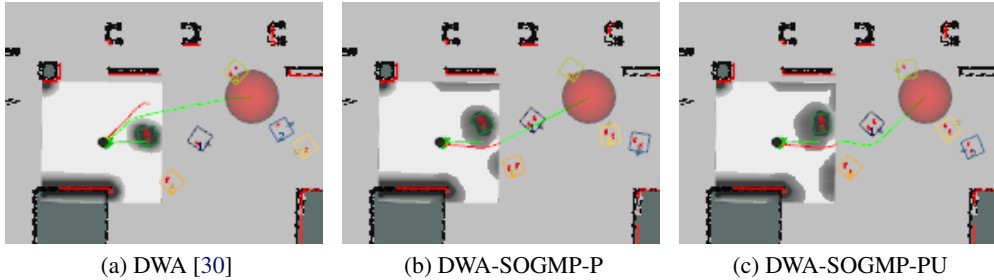

| (a) DWA [30] | (b) DWA-SOGMP-P | (c) DWA-SOGMP-PU |

Figure 14: Robot reactions and their corresponding costmaps generated by different control policies in the simulated lobby environment. The robot (black disk) is avoiding pedestrians (colorful square boxes) and reaching the goal (red disk) according to the nominal path (green line) planned by the costmap (square white map).

