# OpenReview forum: "Stochastic Occupancy Grid Map Prediction in Dynamic Scenes"
_robot-learning.org/CoRL/2023/Conference — CoRL 2023 Poster_

### Official Review · Reviewer_M8W2 · 2023-06-28

**Confidence:** 4
**Originality:** Good
**Technical Quality:** Good
**Clarity Of Presentation:** Good
**Impact:** 3

**Recommendation:**

Weak Accept: I recommend accepting the paper, but will not argue for my recommendation if the majority of other reviewers have a different opinion.

**Review:**

I think that only ego-centric approaches are considered, however, allo-centric occupancy grids also improve the prediction performance, as shown in [1].

I don't agree with the assumption that Eqn.2 does not represent the kinematics and dynamics of the scene. Indeed, this function assumes that the kinematics of the environment can be learned stochastically from sufficient large samples.

In SOGMP, wouldn't it be equal to SOGMP++ with dynamic objects having a velocity of zero? The reason of the performance difference in between these two should be stated more clearly.

The details of the training is not clear to me. In auto-regressive mechanism, do you use the ground truth for each step (t+1, t+2, t+3, ..., t+n) and find the loss at each time step, or do you make the estimations for all n steps (t+n), and compute the loss at the end of n steps only?

For high velocity models (robots or autonomous vehicles), wouldn't a constant velocity motion model assumption fail frequently? Can you analyse your results for a platform for varying speeds?

Would it be possible to give the times/fps rates for the computation power?

For the simulation, usage of the pedestrians does not make any difference, since there are no other types of obstacles. Simple cylinders should also be used, which would accelerate the simulation speed.

[1] Asghar, R., Rummelhard, L., Spalanzani, A., & Laugier, C. (2022, December). Allo-centric Occupancy Grid Prediction for Urban Traffic Scene Using Video Prediction Networks. In 2022 17th International Conference on Control, Automation, Robotics and Vision (ICARCV) (pp. 255-260). IEEE.

**Quality Of The Limitations Section:**

Limitations are addressed clearly

**Questions For Rebuttal:**

- In SOGMP, wouldn't it be equal to SOGMP++ with dynamic objects having a velocity of zero? The reason of the performance difference in between these two should be stated more clearly.
- The details of the training is not clear to me. In auto-regressive mechanism, do you use the ground truth for each step (t+1, t+2, t+3, ..., t+n) and find the loss at each time step, or do you make the estimations for all n steps (t+n), and compute the loss at the end of n steps only?
- For high velocity models (robots or autonomous vehicles), wouldn't a constant velocity motion model assumption fail frequently? Can you analyse your results for a platform for varying speeds?
- Would it be possible to give the times/fps rates for the computation power?

**Robotics Focus:**

Sufficient demonstration on hardware

**Summary Of Paper:**

The paper proposes a novel stochastic prediction algorithm that uses a variational autoencoder to predict a range of possible future states of the occupancy grid. The algorithm takes full advantage of the motion of the robot itself, the motion of dynamic objects, and the geometry of static objects. Three simulated and real-world datasets collected by different robot models are used. A predictive uncertainty-aware planner is proposed to demonstrate the effectiveness of the proposed predictor in simulation and real-world navigation experiments.

**Summary Of Recommendation:**

My main concern is related with the assumption of constant velocity, which over-simplifies the problem. Therefore, more complex cases where the objects don't have constant velocity should be investigated (especially on simulation, since the velocity profiles can be adjusted).

---

### Official Review · Reviewer_Kpfb · 2023-07-12

**Confidence:** 3
**Originality:** Good
**Technical Quality:** Very Good
**Clarity Of Presentation:** Very Good
**Impact:** 3

**Recommendation:**

Weak Accept: I recommend accepting the paper, but will not argue for my recommendation if the majority of other reviewers have a different opinion.

**Review:**

The manuscript is well written and easy to follow. The prediction of future states of scenes for navigation is generally relevant for robotics. The attached video with the navigation experiment helps to judge the applicability of this work.

My main concerns with the manuscript are:
1. The significance of the chosen OGM method for navigation has not been discussed. It is unclear if any of the other baseline OGM methods would achieve a similar navigation performance.
2. The grid resolution (0.1 m) and history length (10) appear low. While experiments show that this is sufficient for navigating with a smaller robot (Turtlebot2) in simulation and a real scenario, the impact of choosing those has not been discussed.

Further, some of the OGM baseline methods are quite old (2 of the 3 are from 2015 and 2016 respectively). While the relation of components in Figure 2 with equations 3 is described in the text, it would help a lot if equations or sections could be referenced directly in the figure.

**Quality Of The Limitations Section:**

Additional details required

**Questions For Rebuttal:**

- 'u' is commonly used for the control input. Is 'u' here the ideal commanded target velocity or the actual measure velocity?
- While the Turtlebot2 and Jackal are skid-steering platforms where the 2D motion model with 2D position with yaw/heading can be applied, the Spot has 6 DoF where this motion model cannot be applied just like that. How is the 6DoF motion mapped onto the 2D motion model and how does this deal with lateral or vertical movement?
- How do the runtime and prediction performance change with higher and lower grid map resolutions and history lengths?
- It is unclear if the OGM evaluation in Section 3.1 runs in realtime (i.e. at 2.95 Hz for ConvLSTM, 5.3 Hz for DeepTracking, 10.68 Hz for SOGMP++, ...) on the Jetson TX2 or offline frame-by-frame. Given the fixed history size (τ=10), wouldn't methods with a slower updated rate, e.g. ConvLSTM, have a longer history (1/2.95 Hz * τ = 3.39 s), and therefore better data, than the proposed faster method SOGMP (1/23.29 Hz * τ = 0.43 s)?
- The navigation experiments only evaluate the proposed OGM method. How would the navigation perform when using the baseline OGM methods in place of SOGMP?

**Robotics Focus:**

Sufficient demonstration on hardware

**Summary Of Paper:**

The work proposes a method to predict the future state of an occupancy grid map (OGM) for robot navigation in 2D. The core contribution of the work is the use of the motion of the robot and dynamic obstacles in its surrounding via ego-motion compensation. The result of the OGM prediction is evaluated on three different platforms, using one simulated scene and two real scenarios. Further, the authors integrate the predicted OGM into a planner and demonstrate this in a real-world navigation scenario.

**Summary Of Recommendation:**

Some properties, such as the impact of choosing a grid resolution and an alternative OGM method for navigation, have not been discussed and it is therefore difficult to conclusively judge the impact of the work. However, the paper has been well written and clearly shows an advantage of incorporating ego-motion into the prediction of future OGM states. The navigation experiment on the Turtlebot2 helped a lot to assess the applicability of the method in a realistic scenario.

---

### Official Review · Reviewer_iQee · 2023-07-19

**Confidence:** 4
**Originality:** Good
**Technical Quality:** Fair
**Clarity Of Presentation:** Good
**Impact:** 4

**Recommendation:**

Weak Reject: I recommend rejecting the paper, but will not argue for my recommendation if the majority of other reviewers have a different opinion.

**Review:**

Strengths:
* The paper is very well written and structured.
* The topic is timely and interesting. The problem is well-motivated and the literature review does a good job of contextualizing the paper in prior work.
* The mapping in Sec. 2.6 from the proposed approach to a Bayes filter was insightful.
* The evaluation in the loop with a planner strengthens the message of the paper.
* The analysis in the appendix provides further context for the method and the claims made in the main paper.

Weaknesses:

* The literature review is missing the following line of work:

[A] T, Hugues, et al. "The Foreseeable Future: Self-Supervised Learning to Predict Dynamic Scenes for Indoor Navigation." arXiv, 2022.

[B] T. Hugues, et al. "Learning Spatiotemporal Occupancy Grid Maps for Lifelong Navigation in Dynamic Scenes." ICRA, 2022.

[C] B. Lange et al. "LOPR: Latent Occupancy PRediction using Generative Models." arXiv, 2022.

* The current use of binary 0/1 OGMs means that occlusions are not accounted for and some probabilistic elements are ignored as well.
* Fig. 1 could be made more informative by creating a color/numbering scheme to track different people through the different stages of the OGM prediction process.
* The size of the considered OGM is quite small (6.4x6.4 m). Prior work looked at bigger OGMs (40-50 m per side). However, the current setting is looking at a small mobile robot rather than an AV, so a smaller OGM may be acceptable. Could you provide a citation for why this is a reasonable size?
* A major contribution in the paper is the VAE architecture that allows for "a range of possible and reliable OGM predictions". However, as far as I could tell, there were no experiments showing the diverse predictions generated by the prediction network. At a minimum, qualitative diverse samples should be presented that show diverse but reasonable potential future predictions.
* It seems that only the simulation data is used for training, unless I misunderstood something. Generally, evaluating human behavior prediction in simulation is suboptimal as simulators provide much cleaner, simpler behaviors than the real-world. The real-world results do seem to indicate reasonable transfer.
* One of my biggest concerns with the paper is that there are no modern OGM prediction methods considered as baselines. For example, [8, 11, A-C] seem like relevant methods to compare against.
* From the quantitative results in Table 1, there does not seem to be a significant difference between the uncertainty and no uncertainty planning performance.
* The claim that moving objects disappear solely due to a lack of stochastic modeling is not entirely true. Disappearance also occurs due to less expressive architecture choices as found by [8, 11]. Potentially, considering more expressive baselines would at least in part circumvent this issue.

The following is a list of minor typos as well as consistency issues I found:
1. What is the github link at the end of the abstract?
2. Line 37: should be 'DOGMs' and in previous work 'DOGMas' [7].
3. The references have some consistency issues. For example, conferences should be capitalized (e.g., [12]) acronyms are sometimes included (e.g., ICRA in [1]) and sometimes not (e.g., [2]). When possible, the conference venue should be cited instead of ArXiv (e.g., [13]). The references should in general be proofread (e.g., [7] lists the year and IEEE twice, LSTM is not capitalized (e.g., [12])).

**Quality Of The Limitations Section:**

Limitations are addressed clearly

**Questions For Rebuttal:**

In addition to the weaknesses listed above, I am hoping the authors can address the following clarifications during the rebuttal period.
1. I think I missed the history and the prediction time horizons. Could you clarify this/point me to where in the paper it is stated?
2. The paper claims that previous observation-based approaches do not account for robot motion compensation. However, the assumption in those methods is that the robot motion is compensated for by the neural network relative to the local frame of reference. Then the claim here becomes that it is easier for the neural network to work with a predefined structure for the robot motion. But the chosen predefined structure seems pretty straightforward (linear). Do you think a more expressive/powerful neural network architecture would circumvent for the robot motion compensation? Or is there a fundamental limitation that cannot be bridged?
3. Could you clarify what is meant by the statement in line 124: "mitigate [the robot's] dynamics effects"?
4. Are $\hat{o}_{t+1}$ only dynamic objects? How are they identified? By filtering them out for the static map?
5. The argument in Sec. 2.5 in favor of the separate static map component states that static objects take up a large proportion of the occupied grid cells. Why would weighting the cross-entropy loss to account for the data imbalance be insufficient?
6. Is the local static map a local map? Is its size also 6.4x6.4 m?
7. Should the prior in the VAE be conditioned on the history of observations? See [D], for example.

[D] M. Itkina, et al. "Multi-agent variational occlusion inference using people as sensors." ICRA, 2022.

8. Do you have a sense for why the WMSE is better without the static map (Fig. 3)?
9. In the video, there is a lot of shaking in the proposed method as compared to the baselines. Why do you think that is?

**Robotics Focus:**

Sufficient demonstration on hardware

**Summary Of Paper:**

The paper presents an approach to occupancy grid map (OGM) prediction that extends prior work by (1) generating stochastic prediction using a VAE and (2) compensating for ego robot motion. The method is evaluated on a collected simulated dataset for simulated crowd navigation, SCAND-Jackal and SCAND-Spot, as well as in the real-world on a TurtleBot robot platform in a real-world setting.

**Summary Of Recommendation:**

Overall, the paper is well written and presents a relevant extension to OGM prediction using generative models. The real-world evaluation is appreciated. My biggest concerns with the paper center around the empirical evaluation. Overall, it is not really completely clear that the uncertainty component helps the planner. Majorly, the lack of stochastic generated prediction qualitative evaluation and the lack of SOTA baseline comparisons for OGM prediction weaken the message of the paper. As it stands, I am leaning towards rejecting the paper at this time. That being said, I think this is a promising paper, and with some improvements to the evaluation it will make a good contribution.

---

### Official Review · Reviewer_QuXe · 2023-07-19

**Confidence:** 4
**Originality:** Good
**Technical Quality:** Good
**Clarity Of Presentation:** Good
**Impact:** 3

**Recommendation:**

Weak Accept: I recommend accepting the paper, but will not argue for my recommendation if the majority of other reviewers have a different opinion.

**Review:**

Review summary:
The paper is well motivated, and the proposed solution is clearly described. The probabilistic problem formulation is very clear, and the simulation results are compelling, using an off-the shelf navigation planner. The real-world robot video is also very compelling. However, the proposed algorithm could be a bit better explained, and some additional qualitative results would help.

Detailed comments:
* Shouldn't the LIDAR conversion function depend on the robot state? Specifically, in eq. 2, shouldn't $c(y_{t-\tau:t})$ be $c(y_{t-\tau:t}, x_{t-\tau:t}, u_{t-\tau:t} )$?
* The functions in eq.3 could be better explained. What exactly is $h(\cdot)$ doing? Is it a learned function, or a geometric operation? Is $k$ just a coordinate transformation? It would help to relate the math notation to the system diagram in fig. 2.
* The paper claims that the primary contribution is the use of the kinematics and dynamics of the robot. The ablation studies seem to support this, but a qualitative analysis of a few test cases showing the difference that this makes would be good. This would help provide an intuition of what kinds of cases this would make a significant difference.
* There is a related field on prediction of tracks of moving objects - while the representation of this problem is different, they both solve the same problem. It would be good to include this in the related work, and ideally also include them in the comparison. Here are two recent example papers from the field:
Kamenev, Alexey, et al. "Predictionnet: Real-time joint probabilistic traffic prediction for planning, control, and simulation." 2022 International Conference on Robotics and Automation (ICRA). IEEE, 2022.
Chen, Yuxiao, Boris Ivanovic, and Marco Pavone. "Scept: Scene-consistent, policy-based trajectory predictions for planning." Proceedings of the IEEE/CVF Conference on Computer Vision and Pattern Recognition. 2022.

**Quality Of The Limitations Section:**

Limitations are addressed clearly

**Questions For Rebuttal:**

It would be good to clarify the math notation questions above.

**Robotics Focus:**

Sufficient demonstration on hardware

**Summary Of Paper:**

This paper tackles the problem of probabilistic prediction of local maps in the presence of dynamic objects around a robot. It formulates the problem as a conditional inference problem, with a sliding window history of observations and robot states. The conditional prediction is performed using a variational autoencoder.

**Summary Of Recommendation:**

Overall the paper presents a reasonable approach (ego-motion compensation + VAEs) to predicting dynamic local maps. The results are good, but the approach could be a bit better explained, and the related work coverage could be expanded.

---

### Decision · Program_Chairs · 2023-08-30

**Decision:**

Accept (Poster)

**Comment:**

The paper presents an approach to occupancy grid map (OGM) prediction that generates stochastic predictions using a VAE and by compensating for ego robot motion. Reviews all agreed that the problem formulation was interesting and principled, with clear results, both in simulation and in extensive real world experiments using a turtlebot/jackal/Spot robot. The primary reviewer concerns were around a lack of modern neural network baselines for occupancy grid prediction and the simple dynamics model used (constant velocity).

The authors responded well, addressing a number of concerns, but did not directly address issues around baselines, instead arguing that modern baselines were infeasible for use here given computational and training requirements. They also argued that an improved dynamics model could be used if desired. Despite these weaknesses, I think the simplicity of the proposed approach and extensive experimental results marginally outweigh these concerns, and recommend acceptance. I encourage the reviewers to address remaining reviewer concerns and incorporate rebuttal clarifications in the final version of the paper.